# Assessing the Environmental Impact of a University Sport Event: The Case of the 75th Italian National University Championships

Lidia Piccerillo *, Francesco Misiti and Simone Digennaro *

Department of Human Sciences, Society and Health, University of Cassino and Southern Lazio, 03043 Cassino, Italy
* Correspondence: lidia.piccerillo@unicas.it (L.P.); s.digennaro@unicas.it (S.D.)

**Abstract:** In recent years, there has been an increase in the attention towards sustainability by the organizing committees of mega-sport events due to the potential impacts they can have. Less attention was given to small-sport events: the carbon footprint of this type of event was seldom investigated unless it had a clear impact on the environment. The purpose of this study is to provide a qualitative-quantitative assessment of the sustainability of a university sporting event held at the University of Cassino and Southern Lazio in 2022. Athletes, technicians, coaches, and companions from various Italian universities were interviewed with the scope being to obtain information on all of the activities associated with the sporting event (travelling, accommodation, etc.). In addition, in-depth interviews were conducted with key members of the organizing committee with the goal of obtaining information relating to the actions undertaken by the Federation to safeguard the environment. The results on the sample indicate a carbon footprint of 40,551 kg of $CO_{2e}$, of which 27,360 kg of $CO_{2e}$ are attributable to transport and 13,191 kg of $CO_{2e}$ are attributable to accommodations. Sporting event organizing committees should implement some strategies to encourage environmentally friendly behaviours to reduce the negative effects of their activities on the environment.

**Keywords:** sport events; sustainability; carbon footprint; $CO_{2e}$ emissions; environmental impact

## 1. Introduction

On 25 September 2015, the United Nations General Assembly (UNGA) adopted the document called: "Transforming Our World: The 2030 Agenda for Sustainable Development." Although the 17 Sustainable Development Goals (SDGs) and 169 related goals do not mention sports, Paragraph 37 of the 2030 Agenda for Sustainable Development recognizes sports as an important factor in sustainable development [1]. In recent decades, the organization of sporting events has significantly increased. Thus, it is necessary that those involved in organizing and participating increase their awareness on the potential effects of such events on the environment. A growing body of literature highlights the negative influence that sportive activities and, in particular, mega sports events can have on the environment [2]. This is the result, among other things, of a greater awareness of climate change, with more attention being paid to environmental safeguarding. In this light, The organizers of the London 2012 Olympics teamed up with British Standards, the U.K.'s National Standards Body, to develop an event management system including event sustainability objectives and sustainable development principles, subsequently called ISO20121 standard for the Management of Sustainable Events [3]. The Fédération Internationale de Football Association (FIFA) and International Ice Hockey Federation (IIHF) have also implemented social and environmental sustainability initiatives via a specific Sustainability & Diversity Department (FIFA) or Environmental & Social Activities Committee (IIHF) [4]. Broadly, Green Sports Alliance, Sports for Climate Action, Sport and Sustainability International, and the United Nations, Sport and the Environment are some examples of sports-specific environmental organizations [5].

The potential environmental impacts caused by sporting events depend on numerous variables, such as the type of sport involved, the size of the event (local, regional, national,

international), the location, the duration of the event, etc. Even the location of the infrastructure, i.e. stadium, airport, and facility-establishment steps, have environmental effects [2,6,7].

For what concerns the size of the event, the following classifications should be taken into account:

- Mega sporting events (for example, the World Cup or the Olympics) are those that produce high levels of tourism, prestige and impact for the community, host city or organization due to their size [8].
- Hallmark sports events (for example, the Tour de France or the Wimbledon tournament) are events with the primary function of increasing the awareness, attractiveness and profitability of a tourist destination [9].
- Small-scale/community sports events are more local or regional events, the impact of which can contribute socially to the local community [10].

The scientific literature shows a certain lack of homogeneity in the methods and indicators for assessing the sustainability of a sports event. One of the first studies in this area was conducted by Collins and colleagues [11], who used two different approaches to assess the environmental impact of "The 2003/04 FA Cup Final" and the "U.K. Stages of the 2007 Tour de France." The study adopted the Environmental Input–Output analysis (ENVIO), which, through visitor spending, is converted into carbon emissions using input-output frameworks. This method allowed a simultaneous assessment of the environmental and economic impact of an event, using similar data and primary survey sources, and with the advantage of allowing to associate economic benefits with selected environmental costs [12]. According to the Ecological Footprint concept, there is a limited amount of biologically productive sea and land to meet all human demands. This concept, expressed in terms of comparable equivalent land units, i.e., global hectares (gha), quantifies the demand that human places on bioproductive areas [11,13]. Among the indicators for assessing the impact of an event, one of the most popular is the carbon footprint, i.e., "the total set of greenhouse gas (GHG) emissions caused directly or indirectly by an individual, an organization, an event, or a product" [14,15]. Emissions of greenhouse gasses other than carbon dioxide are normalized to the global warming potential of carbon dioxide and converted into carbon dioxide equivalents ($CO_{2e}$), which is the carbon footprint's unit of measurement [16]. A few sports organizations [17] adopted the Triple Bottom Line (TBL) concept to measure sustainability success and outcomes in three areas: economic, environmental, and social. Based on the SDGs, Hugaert [18] has developed a new research tool to assess sustainability. Sixteen of the seventeen objectives were converted into items applicable to PSEs, classified according to the three dimensions of sustainability (economic, social, and environmental) and measured on a five-point Likert scale, ranging from strongly disagree to strongly agree.

Sustainable sporting events that minimize negative impacts do not arise by chance. They require a careful process of analysis, planning, and implementation. The impact of small sporting events is probably smaller than mega sporting events, but small events are organized more frequently, so they lead to a large impact that should not be taken lightly [19]. In this light, the paper focuses on a case study related to a small-scale/community sports event: the "75th Italian National University Championships (INUC)," held in Cassino from the 13th to the 22nd May 2022. The GHG emissions associated with travel and accommodations during the 10 days of sports competitions were analyzed. In particular, the study sought to address the following main research questions: (1) Considering the impact of the transport and accommodations of the participants in the 75th NUC, what are the total and individual carbon footprints? (2) Which of the two estimated categories contributes the most to the impact? (3) Which mode of transport used to reach the city of the event contributes the most? (4) What are the perception levels of the key people involved in the organization of the event regarding environment safeguarding?

## 2. Material and Methods

### 2.1. Case-Study: National University Championships Cassino 2022

This case study focused on the Italian National University Championships (INUC), a multi-sport competition that has been organized every year in a different location since 1947. All students enrolled in any degree course of a university recognized by the Ministry of Education in Italy can take part. The INUC is held in two different sessions: the winter session, for snow sports, and the spring session, for the other disciplines. Each year, they are hosted in a different city. The INUC is promoted by the Italian University Sports Centre (IUSC) and organized in collaboration with various University Sports Centres (USCs). Each university has a USC that deals with sports in the university environment. Therefore, with the management of sports facilities owned by the universities, they organize tournaments and courses there and set up representatives that participate in federal championships in the INUC. The 75th edition of this sporting competition was hosted in Cassino from 13th to 22nd May 2022 and was attended by about 2000 athletes-students from all regions of Italy who competed in various sports, including Taekwondo, judo, karate, women's and men's volleyball, tennis, rugby sevens, fencing, futsal, soccer, basketball, athletics, and wrestling. The competitions took place at the sports facilities located in the municipality of Cassino, such as the Gino Salveti stadium and in the neighbouring municipalities.

### 2.2. Assessment Methods

A quantitative method involving the calculation of the carbon footprint was used to answer the first three research questions, while a qualitative method involving a semi-structured interview was adopted to answer the fourth research question.

Carbon footprint analysis requires the definition of three boundaries: the temporal one, the organizational one, and the operational one [20]. The definition of the temporal dimension is necessary to indicate the period over which the emissions were estimated [9]. In this research, the temporal boundary is limited to the days of sporting competition, the organizational one is defined by the event taken into consideration, therefore the NUC, while the operational boundary refers to the selection of the emissions that will be taken into account. In terms of operational boundaries, the researchers distinguish between three different emission areas: Scope 1 (direct emissions), Scope 2 (indirect emissions), and Scope 3 (other optional, in-direct emissions) [14]. This paper focuses on indirect emissions from transport and accommodation. The development of the research tools involved preliminary informal conversions conducted with stakeholders involved in the organization of small sporting events, and the results obtained from the literature review represented the basis for the construction of the questionnaire and the interviews. The data collection took place through a questionnaire administered to a random sample during the 10 days of sports competitions. The sample is made up of players, technicians, coaches, and managers of the USC for a total of 635 respondents. The non-response rate is minimized because the questionnaire was administered by the researchers in person (without affecting the response). Respondents were informed about the research's objective, that participation was voluntary, and that their data would be used for scientific purposes. Participants were asked to answer the following questions: "The following questionnaire is filled in by: (Athlete, Manager, Companion, Coach/Technician)"; "Where did you leave from?"; "University Sports Center of belonging?"; "What mode of transport did you use?"; "Did you come alone or accompanied? If accompanied, by how many people?"; "How many nights do you plan to stay?"; "Where are you staying?"

In the semi-structured interview, a series of previously elaborated open questions are used, as a guide, by the interviewees so the authors structured a series of questions that had sustainability as a theme within the sports organization [21]. Data collection was undertaken via three semi-structured interviews, which were conducted between the 13 and 22 of May 2022. To obtain the most reliable information possible, the authors identified key people who are directly involved in the organization of the Federation. Respondents were employed as an executive, president, or employee of three different University Sports Centres.

Interviewees were also informed of the purposes of the interview and asked for permission to record the interview. The authors transcribed the interviews and, subsequently, the transcripts were analyzed to highlight the findings of the research question.

### 2.2.1. Transport

Since transport is a critical factor in the behaviour of residents and visitors, it is essential to assess the amount of $CO_{2e}$ generated in relation to the travel behaviours of participants [22]. For some modalities, an average emission value obtained from two different methods, highlighted in the following sections, was used. The use of the two methods in combination was expected to provide more extensive results and lead to a more reliable evaluation.

### Road Transport

The $CO_{2e}$ related to road transport was calculated using two methods: one involves the use of the database provided by the Italian Institute for Environmental Protection and Research (ISPRA), and the other one uses the software EcoPassenger calculation [23,24]. Travel-related carbon emissions were estimated based on the distances travelled and the mode of transport to reach the city of Cassino. The data obtained from the questionnaire, on the city of origin, allowed for the calculation of the mileage travelled through Google Maps, assuming that the shortest journey was covered. The travel distances with the respective modes of transport were converted into carbon dioxide equivalent emissions ($CO_{2e}$) using the emission factor table provided by ISPRA. This database is based on the emissions carried out to draw up the national inventory of emissions into the atmosphere. The estimates are drawn up based on national data regarding the fleet and the circulation of vehicles (fleet size, average distances and consumption, speed by vehicle category with reference to urban, suburban and motorway driving cycles, and other specific national parameters). This database is created by ISPRA as an instrument for verifying the commitments undertaken at the international level on the protection of the atmospheric environment, such as the Framework Convention on Climate Change (UNFCCC), the Kyoto Protocol, the Geneva Convention on transboundary air pollution (UNECE-CLRTAP), and the European Directives on the limitation of emissions. The methodology developed and applied to the estimation of atmospheric pollutant emissions is consistent with the IPCC 2006 Guidelines relating to greenhouse gases [23]. The data are updated annually with a delay of about 20 months compared to the end of the year [25] and are available online in Excel format. Therefore, the data referred to in this document are those relating to 2020. At the EU level, polluting emissions from road vehicles are regulated according to the distinction between light vehicles (cars and light commercial vehicles) and heavy vehicles (trucks and buses) [26]. Although ISPRA's data are available and detailed by vehicle engine technology (Euro classes), these are not relevant for calculating $CO_{2e}$, which depends more on the displacement and weight of the vehicle [25]. For car transport, the average conversion factor relating to the auto category was used: for minivans, that relating to the light commercial vehicle category; and for buses and coaches, that relating to the bus category, assuming an average degree of occupancy equal to 50 as indicated by Caserini [25] (Table 1). Based on the equation proposed by Dolf and Teehan [19], the following equation was adopted for the study:

$$i = \left( \frac{d * EF}{O} \right) \tag{1}$$

where $i$ is the impact in kg carbon dioxide equivalents ($CO_{2e}$); $d$ is the distance in km; $EF$ is the emission factor per vehicle for relative travel mode; and $O$ is the number of occupants of the vehicle.

**Table 1.** Emissions factors for the vehicle.

| Mode | EF (kg $CO_2$) |
|---|---|
| Car | 0.163 |
| Buses | 0.0145 [1] |
| Minivan | 0.243 |

$CO_2$, carbon dioxide; EF, emission factors. [1] This value has been estimated by dividing by the average degree of occupancy reported by Caserini [25].

EcoPassenger is a tool for comparing energy consumption and emissions of the main modes of transport (road, rail, and air). The basic methodology for environmental calculations is developed by the Institute for Energy and Environmental Research (IFEU) in collaboration with the Union Internationale des Chemins de fer (UIC) [27]. This software uses energy and emission data from "real world" driving cycles, not from the legislative driving cycle [27]. This tool allows for the calculation of $CO_{2e}$ by setting some parameters (Table 2), including vehicle class, vehicle technology, and the number of passengers in the car. In this study, the selected criterion relating to the vehicle class section is the middle class, the engine is Diesel Euro 4, and the No. of passengers in the car is the real data obtained from the questionnaire.

**Table 2.** EcoPassenger settings table.

| Emission Standard | Energy | Size (Euro Market Segment) | Number of Passengers in the Car |
|---|---|---|---|
| Conventional EURO 1–6 | Gasoline Diesel LPG Battery Electric | Compact class (A, B) Medium class (C, small M) Luxury (D-J) | Variation from 1–5 |

Airplane Transport

In relation to air transport, the calculations were carried out in two ways: one involving the use of the conversion factors published by the Department of Environmental food and rural affairs (DEFRA), and the other using the Carbon Emission Calculator software made available by International Civil Aviation Organization (ICAO) [28,29]. ICAO methodology uses data on trip distance, aircraft types, aircraft fuel consumption, and load factors, according to the equation:

$CO_2$ per pax = 3.16 * (total fuel * pax-to-freight factor)/(number of y-seats * pax load factor) [30].

The UK GHG Conversion Factors have been developed as part of the NAEI (National Atmospheric Emissions Inventory) contract. Values for the non-carbon dioxide ($CO_2$) GHGs, methane ($CH_4$), and nitrous oxide ($N_2O$) are presented as $CO_2$ equivalents ($CO_{2e}$) using Global Warming Potential (GWP) factors from the Intergovernmental Panel on Climate Change (IPCC)'s fourth assessment report [31]. Although DEFRA's conversion factors are estimated based on U.K. statistics, their application is extended to a number of other European countries thanks to their free access, regular updates, and annual recalibrations [32].

Rail Transport

Emissions due to rail transport are based on two methods: one involves the use of the Ecopassenger, which allows for the estimation of the emissions at the departure and arrival stations. The Ecopassenger software for the calculation of railway emissions converts data on energy consumption (Wh) and in $CO_{2e}$ [33]. Through the data made available by Eurostat or IEA, EcoPassenger determines the electricity consumption, energy efficiency, and emission factors of electricity supply for rail transport in European countries [33]. The length of the train route between two connected stations is calculated by the line of sight distance, which is extended by 20–30% depending on cases [27].

The other method involves the use of a conversion factor. Caserini and colleagues [25] suggest to calculate the factor on the basis of the electricity consumption data of an average European train for regional, interregional, and long-distance transport reported by the Mobitool platform, estimating an emission factor of electricity consumption (elaborations made with ISPRA data) and assuming an occupancy factor of 31%.

### 2.2.2. Accommodations

Accommodations, according to Rico and colleagues [34], contribute to GHG emissions due to high-intensity energy. The emission from hotels varies by category. For example, in their study, the emissions ranged from values of 3.9 kg $CO_{2e}$ for overnight stays for one-star hotels to 21.9 kg $CO_{2e}$ for five-star hotels. Instead, Gössling [35] reports that the range of emissions varies between 0.1 kg of $CO_{2e}$ and 260 kg of $CO_{2e}$ per guest per night, depending on the type of accommodation. In this study, it was decided to use a value that refers to the Italian context. The value of 14.3 kg of $CO_{2e}$ per room was derived from the Cornell Hotel Sustainability Benchmarking Index, which uses annual data from international hotel companies and a standardized industry methodology [36]. This value, nevertheless, does not distinguish the different categories of accommodation (two stars, three stars, four stars, five stars, bed and breakfast, and serviced apartment). However, even if this value refers to the Italian context, it is in line with Gössling's [37] estimated worldwide value of 13.8 kg of $CO_{2e}$ per room.

### 3. Results

The questionnaire was completed by 635 people. The athletes, managers, technicians, and coaches that were analyzed took part in 12 different sports disciplines, including seven individual sports (fencing, tennis, judo, karate, track and field, all-in and Greco-Romana wrestling, and Taekwondo) and five team sports (volleyball, football, rugby, five-side soccer, and basketball). The sample represents 26.8% and 25% of the population of athletes and coaches/managers, respectively. Table 3 summarizes information on the type of participant. The questionnaire was completed for the most part (79%) by athletes, followed by technicians or coaches (13%), managers (4%), and companions (4%). As shown in Table 4, accounting for the impact of transport and accommodations, this study estimated a carbon footprint of 40,551 kg of $CO_{2e}$, of which 27,360 kg of $CO_{2e}$ are attributable to transport, making it the largest contributor (67.5%), and 13,191 kg of $CO_{2e}$ are attributable to the accommodations. Divided by the total number of players, coaches, managers, and companions, this equates to 63 kg of $CO_{2e}$ per participant (Figure 1).

**Table 3.** Sample results by type of participant.

| Type of Participant | Frequency (n) | Percent (%) |
|---|---|---|
| Athlete | 500 | 79 |
| Companion | 24 | 4 |
| Manager | 26 | 4 |
| Technician/Coach | 85 | 13 |
| Total | 635 | 100 |

**Table 4.** Total carbon emissions in $CO_{2e}$ kg and in % of the 75th National University Championship (NUC) of the sample.

| Category | Kg $CO_{2e}$ | Percent (%) |
|---|---|---|
| Travel | 27,360 | 67.5 |
| Accommodation | 13,191 | 32.5 |
| Total | 40,551 | 100 |

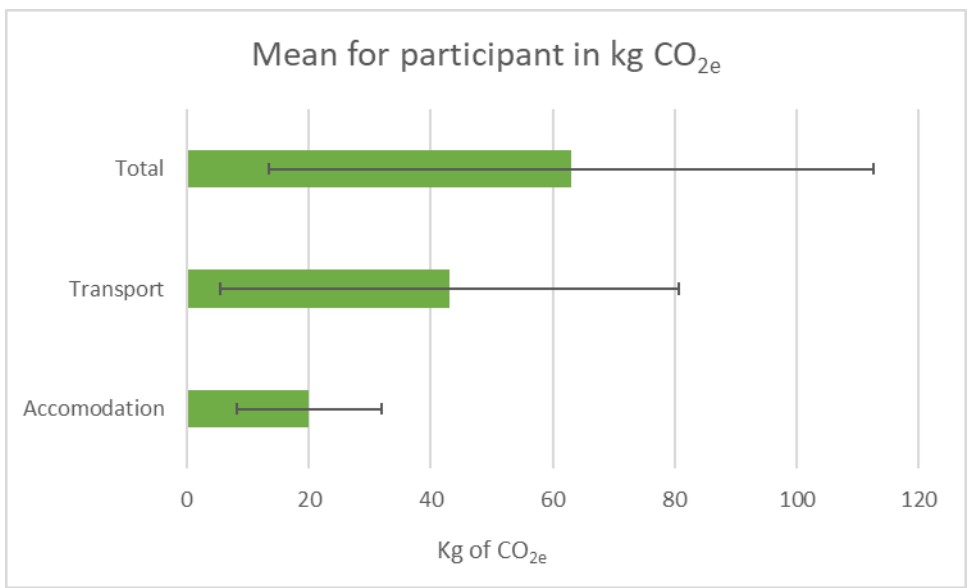

**Figure 1.** Total mean sample of $CO_{2e}$ emissions in kg per participant. (Error bars denote mean ± Standard Deviation).

### 3.1. Results Concerning Road, Airplane and Rail Transport

Based on the methodology presented in Paragraph 2, the carbon footprint attributable to transport (roundtrip) is equal to 27,359 kg of $CO_{2e}$. Travel by minivan (241,440 km) produced 8567 kg of $CO_{2e}$; travel by bus (147,820 km) generated 2142 kg $CO_{2e}$; travel by car (146,994 km) produced 8401 kg of $CO_{2e}$; travel by train (53,123 km) produced 2200 kg of $CO_{2e}$; and travel by airplane produced 6050 kg of $CO_{2e}$ (Figure 2 and Figure 3). Figure 4 compares the distances and carbon footprints for different modes of transportation. The figure indicates that cars and buses both account for 31% of all kilometers travelled yet are responsible for 31% and 8% of the total carbon footprint, respectively. The average emissions by mode of transport varies from a lower value of 14 kg of $CO_{2e}$ for bus transport to a higher average value of 136 kg of $CO_{2e}$ per person emitted for air travel (Figure 5). This result is consistent with what was stated by Montlaur and colleagues [33], according to which short-haul flights generate relatively high levels of emissions and are therefore not very efficient from an environmental point of view. On the other hand, the average number of kilometers travelled between the outward journey and the return journey is more compact, as it varies from 750 km (± SD) traveled by car to 1058 km (± SD) traveled by minivan (Figure 6). Table 5 summarizes the mileage covered and the CO2 emissions and the weight on the total expressed as a percentage for each category of participant, highlighting that most of the emissions are attributable to the athletes. However, given the size of the sample, the highest average is of managers.

**Table 5.** Mileage and $CO_{2e}$ emission by type of participant.

| Type of Participant | Kilometers | Mean (Km) | $CO_{2e}$ (Kg)/(%) | Mean ($CO_{2e}$) |
|---|---|---|---|---|
| Athlete | 51,183 | 1022.4 | 21,188.3 (77.4%) | 42.4 |
| Manager | 23,426 | 901 | 1584.4 (5.8%) | 60.9 |
| Coach | 82,108 | 966 | 3525.4 (12,9%) | 41.5 |
| Companion | 16,890 | 703.8 | 1061.2 (3.9%) | 44.2 |
| Total | 633,607 | 997.8 | 27,359.4 (100%) | 43.1 |

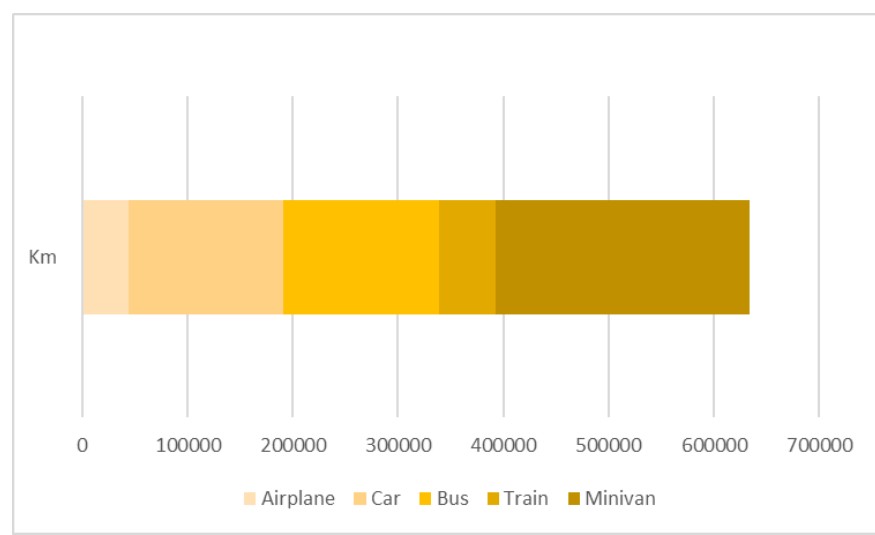

**Figure 2.** Mileage covered with the different modes of transport.

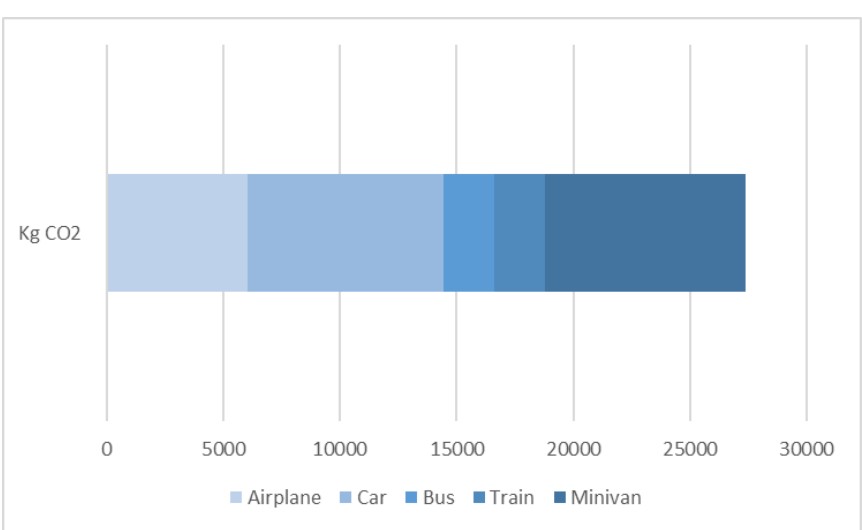

**Figure 3.** Results of $CO_{2e}$ kg of the different modes of transport.

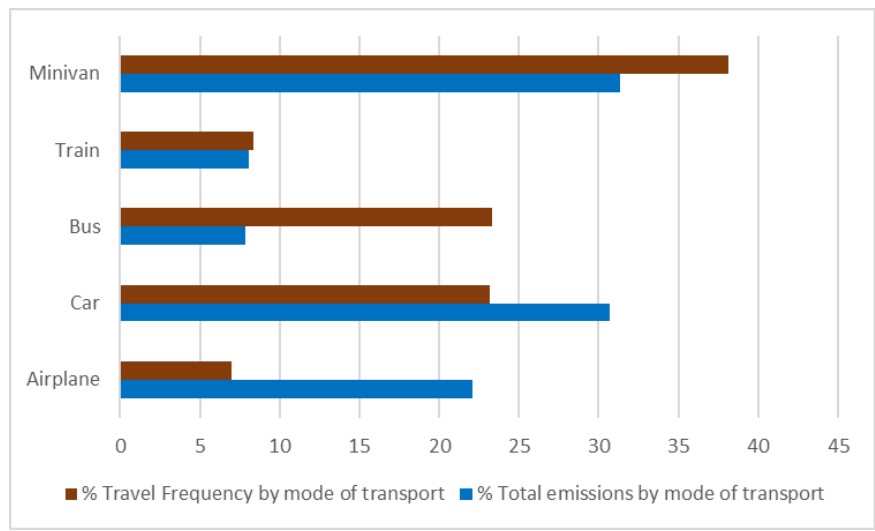

**Figure 4.** Transport mode choices and the transport mode relative contribution to carbon footprint (in %).

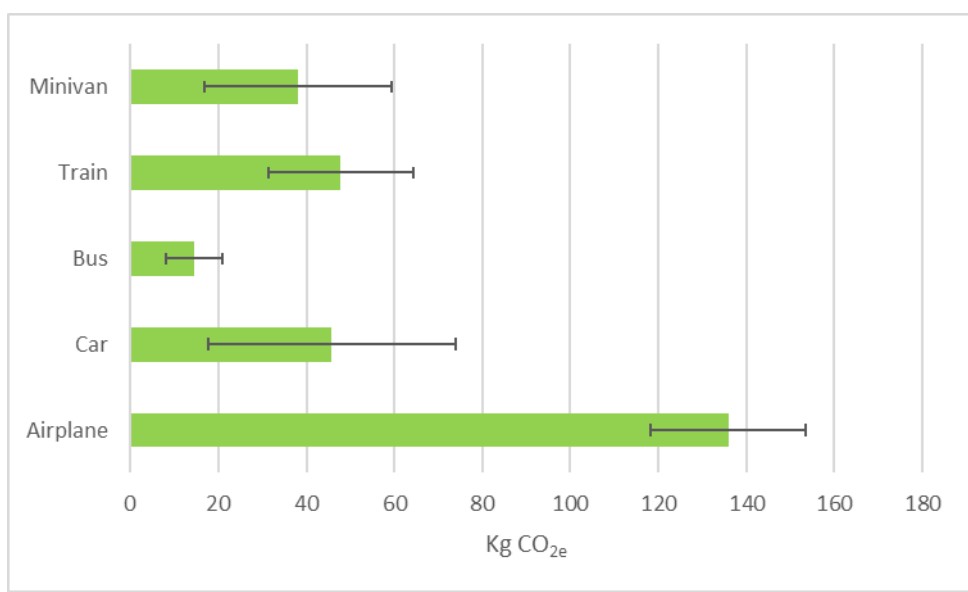

**Figure 5.** Mean of kg $CO_{2e}$ per participant based on transport mode. (Error bars denote mean $\pm$ Standard Deviation).

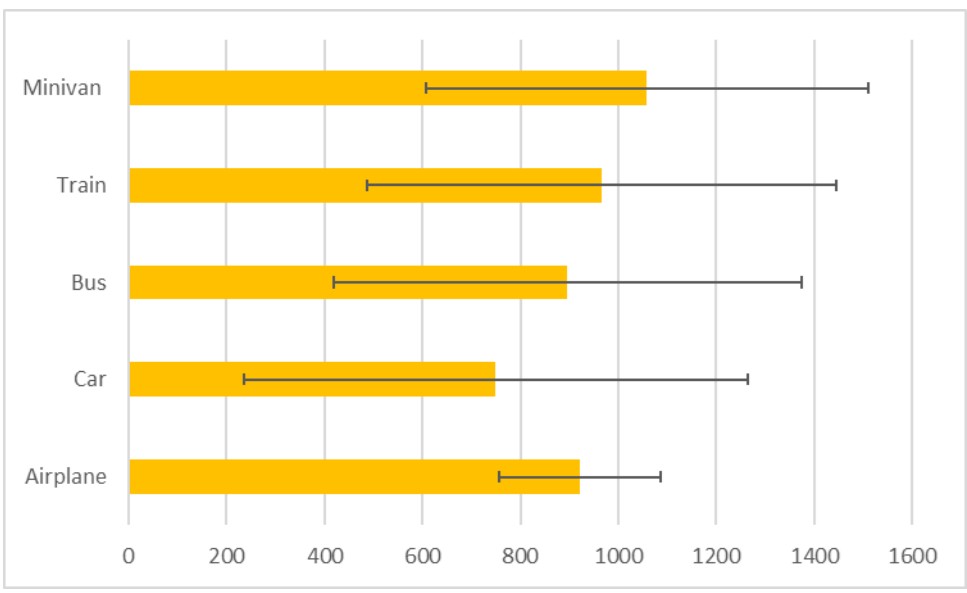

**Figure 6.** Mean kilometers travelled (round trip) per participant by mode of transport. (Error bars denote mean $\pm$ Standard Deviation).

*3.2. Results Concerning Accommodations*

Table 6 summarizes accommodation $CO_{2e}$ by type of participant, highlighting a total of over 13,000 kg of $CO_{2e}$. The detail of these data highlights that, although most of the emissions are attributable to the athletes (80%, as shown in Figure 7) due to the number of the sample of athletes, the highest average of emissions (36.9 kg of $CO_{2e}$) is attributable to the managers as they have a higher average number of overnight stays (5.2 mean of overnight stays) (Table 5).

**Table 6.** $CO_{2e}$ of accommodations by type of participant.

| Type of Participant | Overnight Stays (n) | Mean (Overnight Stays) | $CO_{2e}$ Kg/$CO_2$ % of the Total | Mean ($CO_{2e}$) |
| --- | --- | --- | --- | --- |
| Athlete | 1442 | 2.9 | 10,310.3 (78.2%) | 20.6 |
| Manager | 134 | 5.2 | 958.1 (7.3%) | 36.9 |
| Coach | 243 | 2.9 | 1737.5 (13.2%) | 20.4 |
| Companion | 26 | 1.1 | 185.9 (1.4%) | 7.7 |
| Total | 1845 | 2.9 | 13,191.7 (100%) | 20.7 |

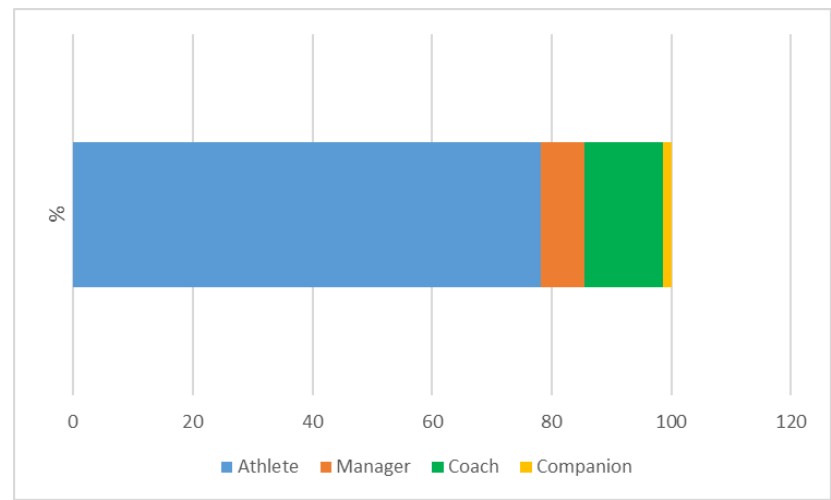

**Figure 7.** Accommodation contribution to carbon footprint by type of participant in percentage.

*3.3. Results Concerning the Interviews*

Interviews were conducted with individuals who have been employed within their respective organizations for more than 20 years as an executive, president, or employee. During the 75th INUC, they were part of the student-athlete accreditation board. It emerged that currently, sustainability is not mentioned in the statute since the last time it was drafted was a period in which this concept was not so taken into consideration and environmental sustainability requirements are not integrated into the processes for sporting events. According to one of the interviewees, environmental sustainability is used more as a secondary tool to obtain investments and economic savings rather than promote environmental benefits. However, for example, some more sustainable activities from an environmental point of view related to the sporting event have been implemented. For example, with the scope to reduce energy and paper consumption, some processes have been made digital, and the number of opening hours of the accreditation phase of the athletes has been reduced. Regarding the INCU, according to the interviewees, the objective of implementing strategies that reduce the impact on the environment is entrusted to the local organization that manages the sporting event from year to year. In conclusion, the analysis of the interviews showed that the key strategic interest is the broadest participation in sports, and that less attention is given to the impact of the events on the environment.

**4. Discussion**

The United Nations Framework Convention on Climate Change Sports for Climate Action (UNFCCC) sets the measurement, reduction, and communication of greenhouse gas emissions as one of the objectives that sports organizations are required to pursue. To this end, the methodology adopted for this study allows for environmental assessments of sport events with easily available data and limited costs [38].

The first research question objective of this study was to calculate the emissions attributable to a small-scale/community sports events. The emissions caused by transport and accommodations were estimated.

The result shows that the estimate of greenhouse gas emissions is equal to 40,551 kg of $CO_{2e}$. The means carbon emissions per person during the 10 days of sporting competitions resulted in 63 kg of $CO_{2e}$, which, divided by the average number of days spent by the sample, turns out to be equal to 21.7 kg of $CO_{2e}$ per day per person. While considering different methodological choices that make comparisons between different sporting events difficult, the average carbon footprint of the participants in this study is similar to the average participation of a team member in a university sporting event [19] and the average number of spectators at the British stages of the 2007 Tour de France [12].

The second research question was to determine which, between accommodation and transportation, caused the most emissions. The findings of this study show that the majority (67.5%) of emissions are attributable to transport and 32.5% are attributable to accommodations (Table 4), in line with the results of other study. For example, in a study of a university event conducted in Arizona, the impact of travel amounts to 77.8%, and that of accommodation to 19.3%, of the total impact [39]. The modes of transport have different impacts: those who used the air mode had an impact of almost 9.7 times more than those who travelled by bus. In terms of percentage, although only 7% of the kilometers were travelled by air, as much as 22% of GHG emissions are attributable to this mode. From the results of this study, it would seem that the mode that causes the lowest emissions is bus transport, consistent with what was reported by Pereira and colleagues [40]. Given the representativeness of the sample, assuming that there are no geographic biases in the sample, with a certain degree of approximation, it can be stated that the total emissions of the event are greater than 148,000 kg of $CO_{2e}$.

From the analysis of the interviews emerged the fact that the Federation is not currently actively engaged in projects that aim to achieve one of the sustainable objectives. However, the promotion of sports as a tool for social inclusion, cultural growth, and consolidation of social relations indirectly contributes to the achievement of the social sustainable development goals. There are a lack of initiatives in terms of environmental sustainability and evaluation of their impact. Although some actions have been implemented, according to what is reported in the literature on strategic sustainable development in sports organizations, these actions appear to be unplanned and fragmented [41]. These results are in line with a recent study on football clubs' commitment to sustainability [42]: environmental issues have not so far been a top priority, but there is a commitment from a social point of view by sport organizations.

This work adds to the body of literature on environmental sustainability in sports since it was, to the best knowledge of the authors, the first study to estimate the carbon footprint of a university sports event in Italy. The findings of this study have practical implications for those involved in the organization of sporting events because often the various stakeholders are unaware of the contribution their actions make to climate change. Understanding the environmental impact could help organizing committees implement strategies to encourage more sustainable behaviours [43]. Although this study did not calculate the emissions related to travel between lodgings and the different sports competition venues, which were located not only in the municipality of the main venue of the event but also in neighbouring municipalities, it can be hypothesized that they further contribute to the total impact on the environment. Thus, it is advisable that local organizing committees of INUCs seek to favour the free use of public transport between the different venues of the sporting event in order to limit the emissions caused by travel by private vehicles or taxis within the city hosting the sporting event. Considering that many emissions are caused by air travel, short domestic flights could be replaced by more efficient ways to travel.

As suggested by Loewen and Wicker, organizing committees could offer a discount for sponsored trains or buses that encourage participants to avoid air travel to reach the hosting city event [16]. Another strategy to reduce GHG emissions could be to improve environmental awareness by educating student-athletes on more sustainable behaviours. In addition, Italian Sports Federation of University Sport could add environmental sustainability objectives and criteria in its statute, mission, and procedures. They could also commit themselves to adopted

guidelines and regulations on environmental matters so that the organizing committees will have to operate in compliance with them. Ultimately, the Federation could consider investing in impact-offsetting projects or carbon-offsetting credits.

Regarding the SDGs, these strategies could be linked not only to Objective 13 (take urgent action to combat climate change and its impacts) but also to Objective 3 (promotion of health and well-being for all), since GHG emissions have consequences on climate change and have a negative impact on the health of the population [44]. Therefore, effective and appropriate strategies can determine public health benefits [45].

Nonetheless, although the sustainability paradigm implies an intrinsic relationship between environmental, economic, and social aspects, this study focused on the environmental one and has some limitations. A limit of this case study is related to the evaluation of the return journey emissions because it assumes that the participants used the same means of transport and that they travelled the same number of kilometers to return home. There may be an underestimation of the emissions caused by the journey as it assumes that the shortest route was taken. Furthermore, as previously mentioned, no data was collected on the travel between the athletes' accommodations and the various venues of sports competitions, so the travel impact component is probably underestimated. In addition, while the study refers to a sample of the total reference population, there is also a lack of data relating to other types of participants in the sporting event, i.e., referees and various match officials and volunteers for whom the actual impact of the event is likely higher than estimated by this study. The emission factor used for accommodations in this study is unique for each category of accommodation. However, the impacts of different categories could be different, in fact. For example, peer-to-peer online accommodation platforms appear to have low emissions [46]. Moreover, this study did not cover the energy consumption emissions related to sports facilities used for competitions and the facility where the accreditation procedure was conducted for almost the entire duration of the competitions.

Future research could consider a specific emission factor for accommodation categories, as the type of accommodation chosen by a participant can influence the impact of this category, and having participants informed about the amount of an impact category over another could help reduce emissions from a sporting event. The methodology of this case study could be extended and provide wider results by collecting information on further activities related to the sporting event by, for example, collecting data on food and drink consumption, waste, distributed gadgets, and energy consumption of sports facilities. Obviously, all this implies the need to create synergies and forge partnerships with all the parties involved in the process, from the Federation to the organizing committee to the host municipality.

## 5. Conclusions

This case study offers insight into the sustainability of a sporting event in a university sports setting from an environmental impact point of view by proposing a qualitative-quantitative method. It measured the environmental impact in terms of the carbon footprint of the 75th National University Championships by estimating the GHG emissions attributable to the transport and accommodations of the participants. It verified which activities contributed the most to emissions and investigated how much the sports organization considered environmental sustainability. Finally, it proposes potential actions that could be taken to reduce emissions. Given the current climate change and considering the necessity to achieve the SDGs, it is of fundamental importance to define protocols and guidelines that might be implemented to promote the sustainability of small-scale sporting events, and these methodologies should highlight the most critical areas, acting as clear and understandable measurement tools even by the less experienced in the field so that they can have a concrete application. It seems appropriate and necessary to encourage debate and interaction between the organizers of sporting events and the scientific community in order to identify, experiment, and promote new methodological and cognitive tools that allow a more careful and aware planning of the impacts on the sustainability of ecosystems.

Finally, it is advisable that the methodology, findings, and recommendations provided by this study will guide committees involved in the management of sporting events towards better environmental management.

**Author Contributions:** Conceptualization, L.P. and S.D.; methodology, L.P., S.D., F.M.; formal analysis, L.P.; investigation, L.P.; writing—original draft preparation, L.P., S.D., F.M.; writing—review and editing, L.P., S.D., F.M.; supervision, S.D., F.M. All authors have read and agreed to the published version of the manuscript.

**Funding:** This research received no external funding.

**Institutional Review Board Statement:** Not applicable.

**Informed Consent Statement:** Informed consent was obtained from all subjects involved in the study.

**Data Availability Statement:** The datasets presented in this study are available on request to the corresponding author.

**Conflicts of Interest:** The authors declare no conflict of interest.

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
