# Peer review of "Assessing the Environmental Impact of a University Sport Event: The Case of the 75th Italian National University Championships"

_sustainability, doi:10.3390/su15032267_

Round 1

Reviewer 1 Report

Please include additional details on the development and implementation of the survey instrument. More information on the distribution of the responses compared to the distribution of the attendees and any other characteristics, e.g., age, sex, of the respondents should be included. It would be helpful to make the survey instrument available as supplemental information. 

Please include additional results by participant type such as use of transportation mode and total (accommodation + transport) CO2e per day.

Please describe how the results scaled from the survey respondents to the number of attendees in order to calculate the carbon footprint of the event.

If the goal is to compare participant types (ref Tables 3 and 5) or transport modes (ref Figure 5), statistical analysis would be helpful.

Typographical errors?

Line 466:

 “...committee their self...” 

may be “...commit themselves...”

Line 477:

“...casa-study...”

may be “...case study...”

Reviewer 2 Report

Thank you for this very clear CO2 assessment article. My first general comment is that you need to elaborate on your sampling methodology and the relationship to the values you are providing. Please clarify how you recruited the 635 respondents from the 2000+ participants. Indicate your non-response rate. If this was a random sample, you should be using inferential statistics to estimate the total CO2 values with a margin of error. If this was not a random sample, indicate possible sources of bias in your sample. If this was not a random sample, you should clearly indicate that the total values given are for a sample and that the actual totals for the event are likely around three times higher, presuming that there is no clear geographic bias in your sample. My second general comment driving the detailed recommendations below is that since most readers of this journal will probably need to have the numbers presented in some kind of relative context so their importance can be assessed. You need to provide clearer context for both your total and per-capita CO2 values. Tables and/or charts showing these values in relation to similar results from other similar studies of similar or contrasting events would be helpful. You reference some of these in your discussion. Can you provide a graph showing the relative CO2 intensity per passenger-km for the different travel modes. I suggest moving your reference to the high CO2 intensity of short-haul air travel into the results section so there is a clearer explanation for the high CO2 intensity of air travel. A bar graph of individual travel distances by mode with stdev whiskers (similar
to figure 5) might clarify the modal choices of the respondents. Finally, if you are going to invoke sustainability discourses, you need
to place this type of event in the context of broader energy usage.
Total Italian CO2 emissions in 2019 were around 317,240 kt (World Bank).
In that context, dozens of annual events of this type would still only
represent an insignificant share of overall emissions. Arguably, the
benefits of this type of event to the associated institutions and to
society at large could be argued to be well worth the minor externalities. Good luck.

Reviewer 3 Report

The paper “SportPrint: Assessing The Environmental Impact of an University Sport Event. The Case of the 75th Italian National University Championships” fully complies with the topic of the issue Special Issue "Sustainability of Festivals and Events". In general, the paper comprehensively covers the issue of organizing sports events on the environment, starting from a solid literature review to an assessment and calculation of CO2e and further to the Discussion chapter. This chapter appropriately discusses findings along with the existing gaps and prospects for implementing sustainability during the organization of sports events. In order to improve the quality of the paper, I propose making the following changes:

Title: The coined word “SportPrint” appears only at the beginning of the title and nowhere else, therefore I propose erasing it. Other vices, it can be kept only if the authors which to further elaborate it in the text e.g. to name a special methodology, but in such a case, it requires substantial changes in the text. Therefore I propose the title: “Assessing the Environmental Impact of a University Sport Event: The Case of the 75th Italian National University Championships”.

Abstract: Please unify “CO2e” or “CO2e” in the abstract and throughout the text including figures.

Lines 25-89 and 90-164: introduction and Literature review

Unfortunately, in the Instructions for Authors of the journal Sustainability in “Research Manuscript Sections” there is no “Literature review” as a separate chapter, but rather it is advised to integrate it into the Introduction. Therefore, I propose to condense the text of the Introduction and incorporate key references from the present Literature review, making the final Introduction the length of the current Introduction. 

Line 254: Write “Minivans” in small letters in the text as other vehicle categories are written. 

Line 264: Please write “(IFEU)” in capital letters. 

Line 267: For “CO2 emissions” please mention the short e.g. “CO2e” at the beginning of the text and apply it to the rest of the text.

Line 268: Explain the term “load factor”.

Line 272: Under table 1 explain the meaning of “EF” please.

Line 334: The title for table 3 is missing. Please add the title. And maybe it would be good to draw a line under the row “Total” to distinguish it from the rest of the rows. The same can be applied to table 4. (Line 375).

Lines 345, 359, 361 and 365: Please delete titles which are inside the figures, since there are already appropriate titles below each figure.

Line 362: Not “Kg” but “kg” and please check the rest of the text.

Line 378: Please add “among the interviewed persons”, since now it seems that all interviewees have 20 years of experience.

Lines 396-503: Each paragraph in the chapter Discussion has its point, but sentences are rather too long. In general, I propose again condensing the text of the completed Discussion to fit around one page.

Line 448-459: Concerning savings in transportation, maybe it would be interesting to separate and to calculate (or just to propose for some next research) CO2e generated during transport engaged specifically for this occasion (e.g. own minivans, busses, charter flights) and using regularly scheduled transportation (e.g. buses, trains, etc.).

Lines 512-514: Instead of repeating problems in methodological approaches, just mention the emission of CO2e generated (i.e. by presenting exact numbers) and mention the chosen method, as well as % of estimated emissions by transportation and accommodation.

******************

Finally, so far English language requires some small corrections (e.g. "a university" not "an university"), but since rewriting some chapters is necessary I suggest to the authors carefully check English upon correcting the paper.

Round 2

Reviewer 2 Report

Good work.

Reviewer 3 Report

Dear authors,

since you have addressed all the item I have suggested I am satisfied with the outcomes. Therefore, I suggested to the Editor to ACCEPT the manuscript for publication.